# Predicting Severity of Acute Pancreatitis

**DOI:** 10.3390/medicina58060787

**Published:** 2022-06-11

**Authors:** Dong Wook Lee, Chang Min Cho

**Affiliations:** Department of Internal Medicine, School of Medicine, Kyungpook National University, Daegu 41944, Korea; storm5333@naver.com

**Keywords:** acute pancreatitis, severity, predicting factors

## Abstract

Acute pancreatitis has a diverse etiology and natural history, and some patients have severe complications with a high risk of mortality. The prediction of the severity of acute pancreatitis should be achieved by a careful ongoing clinical assessment coupled with the use of a multiple-factor scoring system and imaging studies. Over the past 40 years, various scoring systems have been suggested to predict the severity of acute pancreatitis. However, there is no definite and ideal scoring system with a high sensitivity and specificity. The interest in new biological markers and predictive models for identifying severe acute pancreatitis testifies to the continued clinical importance of early severity prediction. Although contrast-enhanced computed tomography (CT) is considered the gold standard for diagnosing pancreatic necrosis, early scanning for the prediction of severity is limited because the full extent of pancreatic necrosis may not develop within the first 48 h of presentation. This article provides an overview of the available scoring systems and biochemical markers for predicting severe acute pancreatitis, with a focus on their characteristics and limitations.

## 1. Introduction

Acute pancreatitis (AP) is the development of acute inflammation in the pancreas and is one of the most common disease entities in gastroenterology that requires hospital admission [1]. AP is diagnosed when two or more of the following three findings are satisfied [1,2]: (1) upper abdominal pain suggestive of pancreatitis, excluding pain from other conditions, such as gastric/duodenal ulcer perforation or aortic dissection, myocardial infarction, etc.; (2) an increased level of serum amylase or lipase exceeding the upper limit of normal values by more than three-fold; (3) findings of image studies, such as ultrasonography, computed tomography (CT) or magnetic resonance imaging (MRI) suggestive of AP. According to the Atlanta classification revised in 2012, AP can be classified into the following categories according to severity: mild, moderately severe, and severe (Table 1). Mild AP is characterized by a lack of organ failure and no local or systemic complications. Moderately severe AP is indicated by transient organ failure (resolves within 48 h) and/or local or systemic complications. Lastly, severe AP is characterized by persistent organ failure that may involve one or more organs [1]. The mortality rate of AP is between 3–10%; however, the mortality rate of severe AP increases to 36–50% [3,4] and if there is no adequate treatment in mild AP, it can progress to severe AP [5]. Therefore, a prompt and accurate prediction and response to severe AP are essential for improving the prognosis of patients. In this review, various factors and scales associated with severity predictions of AP are discussed.

## 2. Predictors Using Patient’s Factors

To predict the severity of AP, the accurate obtaining of details of medical history and physical examination, as well as a review of systems, must be performed. Among the various etiologies that cause AP, alcohol leads to more severe AP and increases the risk of pancreatic necrosis requiring intubation [6,7]. It is argued that hypertriglyceridemia (hyperTG)-induced AP is more severe than AP caused by other etiologies. In a systematic review, the severity of hyperTG-induced AP is compared with AP caused by other etiologies, and six studies showed that hyperTG-induced AP was more severe, while three studies reported no difference in severity according to etiology [8]. However, the criteria for measuring the severity of AP were different in each study, and there were several case series or case–control studies without a randomized controlled trial in the included studies. As such, no definite conclusions have been reached yet. Several studies investigated the relationship between AP and obesity. In a meta-analysis study, severe AP was more frequently observed in patients with a body mass index (BMI) of 30 or higher (odds ratio (OR) 2.3, 95% confidence interval (CI) 1.8–4.6), and they had a significantly greater risk of local and systemic complications (local OR 3.8, 95% CI 2.4–6.6; systemic OR 2.3, 95% CI 1.4–3.8) [9]. In particular, when BMI > 25, the risk of severe AP increases, while mortality does not increase. In contrast, in patients with BMI > 30, both the risk and mortality of severe AP are increased [10]. Additionally, other studies have shown that hypertension increases the risk of renal failure, which leads to a greater risk of severe AP and an increased length of hospitalization [11]. Steatohepatitis is also associated with the risk and severity of AP [12].

AP is closely related with cytokines, and several studies actively investigated the relationship between the severity of AP and genetic factors generating cytokines. GSTM1, GSTT1, GSTP1, CASP7, CASP8, CASP9, CASP10, LTL, TNFRSF1B and TP53 genes are associated with apoptosis and are susceptible to AP [13]. Follow-up studies are currently actively investigating possible individualized treatment and prevention strategies for AP.

## 3. Predictors Using Biochemical Findings

### 3.1. Hematocrit

The third-space loss of intravascular fluid is induced in AP, which can lead to hemoconcentration and is clinically confirmed by an elevated hematocrit level. In studies that used hematocrit to predict the severity of AP, the cut-off value and the time of blood sampling were different for each study. However, studies commonly report that patients with no increased hematocrit at the time of admission, and in the first 24 h after admission, show a better prognosis [14,15].

### 3.2. Blood Urea Nitrogen/Creatinine

In a study of 5819 patients from 69 institutions, mortality increased when blood urea nitrogen (BUN) increased by more than 5 mg/dL within 24 h of hospitalization for AP [16]. Another study reported that mortality increased when the serum level of BUN was greater than 20 mg/dL at the time of hospitalization [17]. BUN reflects the initial condition of the patient and is a useful indicator for the appropriateness of initial resuscitation. In a prospective study of 129 patients with AP, pancreatic necrosis was more frequent in patients with levels of serum creatinine greater than 1.8 mg/dL in blood tests performed within the initial 48 h [18].

### 3.3. C-Reactive Protein

C-reactive protein (CRP) is an acute-phase reactant synthesized in hepatocytes after stimulation by serum interleukin (IL)-1 and IL-6. CRP is a single indicator, commonly used to evaluate the severity of AP. CRP measured 48 h after the onset of symptoms can predict the outcome of AP [19], and CRP (cut-off value 150 mg/L) can be used to distinguish between mild and severe AP with 86% sensitivity and 61% specificity [20,21]. However, its accuracy is low within the first 48 h of symptom onset, and as CRP is synthesized in the liver from serum cytokine actions, it can be underestimated in patients with liver disease due to alcoholism or obesity, which is common in AP [22]. Furthermore, a recent Cochrane review failed to prove the role of CRP in the diagnosis of pancreas necrosis [23]. Most guidelines on AP advise against the use of a single marker to triage patients; however, a CRP level more than 150 mg/dl at 48 h after admission can predict a worse prognosis of AP [24].

### 3.4. Cytokines

Various cytokines are involved in the development of AP and multi-organ failure. Studies attempted to predict the severity of AP using concentrations of different cytokines. In one study, the combination of IL-10 and serum calcium predicted organ failure in a relatively accurate manner (sensitivity 88%, specificity 93%) [25]. Furthermore, there is a meta-analysis about early biomarkers in AP and IL-6 (more than 50 pg/mL) playing a role in the early prediction of progression to moderately severe or severe AP (sensitivity 87%, specificity 88%) [26]. Although other biomarkers such as IL-8 and the tumor necrosis factor α were used to predict the severity of AP, there are some limitations to overcome, such as strong evidence based on large-scale studies and the lack of methods to easily measure these cytokines. Thus, these limitations interfere with the clinical implications of cytokines for predicting AP severity [27,28].

### 3.5. Others

Procalcitonin is detected in the serum of patients with severe bacterial or fungal infections and in patients with multi-organ failure. Several studies report that procalcitonin at the time of hospitalization is a better predictor of AP severity than the Acute Physiology and Chronic Health Examination (APACHE) II score or CRP level [29,30]. In addition, quantitative test results show that trypsinogen-2 in the urine is significantly higher in patients with severe AP [31,32], and measuring trypsinogen activation peptide (TAP) in the urine of patients was helpful in predicting the severity of AP within 24 h of onset [21]. However, there is a lack of follow-up studies to support these findings, and there are no methods to easily measure the levels of these factors in clinical settings.

## 4. Predictors Using Radiologic Image

### 4.1. Computed Tomography

Contrast-enhanced abdominal CT is widely used to assess complications, and the Balthazar grade is the most commonly used score to evaluate AP severity (Table 2) [33]. In addition, CT is useful in evaluating the severity of AP as findings without contrast enhancement indicate pancreatic necrosis [34]. Based on these characteristics, scoring the Balthazar grade and using pancreatic necrosis as a CT severity index has been suggested as an important evaluation method for assessing AP severity (Table 3) [35]. In that study, a CT severity score of 0-3 points was associated with a 3% mortality rate, while a score of 7–10 points increased the mortality rate to 17%. Mortele et al. simplified the CT finding and pancreatic necrosis levels into three grades and introduced a modified CT severity index with extrapancreatic complications (Table 3). This new index has a higher accuracy in predicting hospitalization period, surgical treatment, infection and organ failure [36]. 

Several studies investigated other findings for CT (e.g., pleural effusion, body composition) as predictors of AP. Heller et al. reported that pleural effusion was observed in 84.2% of patients with severe AP, while it was observed in only 8.6% of those with mild AP [37]. Furthermore, the pleural effusion volume was associated with the duration of admission, and CRP levels showed the possibility of a reliable radiologic predictor of the severity of AP [38]. Subcutaneous or visceral adipose tissue, skeletal muscle mass or density, and mean muscle attenuation were evaluated in the context of CT defined body composition analysis, and reported promising results for the prediction of AP severity; however, a lack of prospective studies of different races is hurdle to overcome [39,40].

Although CT is an important test, it has several limitations. The CT severity index has the highest accuracy at 6–10 days from the onset of AP. In particular, necrosis of the pancreas is often not observed in CT images within 48 h of onset. Therefore, CT may not be an adequate method to evaluate the severity of the early phase of AP. Furthermore, possible complications caused by the contrast agents must be considered. Therefore, selective CT examination is recommended.

### 4.2. Others

Trans-abdominal ultrasonography is limited for evaluating pancreatic necrosis. However, as contrast-enhanced ultrasonography was recently developed, studies evaluating the blood supply or necrosis of the pancreas have been reported [41,42]. Endoscopic ultrasonography (EUS) can be useful for observing the structure and parenchymal changes of the pancreas. In particular, EUS is useful in evaluating the etiology of AP, such as biliary AP [43,44], and there is a study showing that radial EUS can be helpful in predicting the severity of biliary AP with a sensitivity of 89.7% and specificity of 84.2% [45]. However, no follow-up studies have demonstrated the superiority of EUS compared with CT, and the role of EUS in predicting AP severity is limited. 

MRI is useful for evaluating not only necrosis, but also local complications, such as bleeding, fluid collection, pseudocysts, abscesses, and pseudoaneurysms [46]. Moreover, peripancreatic vascular changes or the severity of vascular involvement is positively correlated with the severity of AP [47]. However, MRI requires patients’ to co-operate and hold their breath to prevent motion artifacts, and MRI is a time-consuming modality. Thus, there are limitations to practically performing MRI in the clinical field.

## 5. Predictors Using Scoring System

### 5.1. Ranson Score

The Ranson score was developed for the evaluation of the severe status in patients with AP. Alcoholic AP is evaluated using a total of five factors at the time of admission and six factors after 48 h of hospitalization. Biliary AP is assessed using five factors at the time of admission and five factors after 48 h of hospitalization (Table 4) [48,49]. The Ranson score has area under the receiver operating characteristic curves (AUCs) of 0.84, 0.56, 0.80 and 0.810 for the prediction of AP organ failure, necrosis, mortality and severity, respectively. However, as the Ranson score must be evaluated at the time of admission as well as after 48 h of hospitalization, it has limited ability in predicting severity in patients with AP at the time of admission, which leads to delayed initial treatment. 

### 5.2. Glasgow Score

Although the Ranson score allows for a relatively accurate prediction of AP severity, it is complex and has many application items. Therefore, an alternative scoring system, the Glasgow system, was introduced. The Glasgow score can be evaluated within 48 h of hospitalization and measures serum albumin instead of hematocrit, base deficit, and fluid sequestration (Table 5) [50]. Unlike the Ranson score that is applied differently according to the cause of AP, the Glasgow score is universally applied for all causes. It is simple to use and has a similar accuracy to that of the Ranson score, with an AUC of 0.78 for the prediction of AP severity [51].

### 5.3. APACHE II Score

The APACHE II score is not an evaluation for a specific disease. Instead, the APACHE II score is an indicator used to classify patients who need to be treated in an intensive care unit (ICU), and evaluates a total of 12 clinical indicators, including age, underlying diseases, and mentality at admission [52]. In patients with an APACHE II score of less than eight points, mortality was lower than 4%. However, if the APACHE II score was higher than eight points, the mortality rate was between 11 and 18%. The APACHE II score is an effective scale for predicting severe AP; however, this scoring system is complicated and inconvenient. Additionally, the APACHE II score was excessively high in older age groups. 

As obesity is reported as an important factor for predicting the mortality of AP, a new APACHE-O score system was proposed after the addition of BMI to the previous APACHE II score; however, the APACHE-O score did not have an improved accuracy compared to APACHE II [53,54]. 

### 5.4. Bedside Index of Severity in Acute Pancreatitis

The bedside index of severity in acute pancreatitis (BISAP) score predicts severity based on five factors measured 24 h after hospitalization (Table 6). When no factor is satisfied, mortality is less than 1%; however, mortality increases beyond 22% when all five factors are satisfied [55]. In a study of 18,256 patients, the BISAP score had a similar accuracy to APACHE II (BISAP AUC: 0.82, 95% CI 0.79–0.84 versus APACHE II AUC: 0.83, 95% CI 0.80–0.85) [55]. Another prospective study shows that the risk of organ failure and pancreatic necrosis significantly increased to 7.4 (95% CI 2.8-19.5) and 3.8 (95% CI, 1.8–8.5), respectively, when the BISAP score was three points or higher [56]. BISAP can predict severity, organ failure and mortality in AP very well, and to the same level as other scoring systems; therefore, it is proven to be a good predictor, mainly in Western environments [57].

### 5.5. Japanese Severity Score

The Japanese severity score (JSS) was first proposed in 1998 and modified in 2002. This scoring system classifies severity from stage 0 to stage 4, with the total score ranging from 0 to 27 points (Table 7) [58]. JSS has a similar accuracy to the Ranson score in predicting mortality and severity; however, it is complicated and limited in clinical use.

### 5.6. Harmless Acute Pancreatitis Score

The harmless acute pancreatitis score (HAPS) is another system that consists of rebound abdominal tenderness and/or guarding, serum hematocrit and creatinine levels [59]. Thus, it is a simple, reproducible system without complicated tests and can be investigated and interpreted within 1 h of admission. According to HAPS, patients with AP are less likely to progress to severe AP if rebound tenderness is not present and serum creatinine and hematocrit are within the normal range at admission. A prospective validation study reported that HAPS could predict a non-severe disease course with high specificity (97%) and positive predictive value (98%) [59]. In reality, HAPS can be applied in low-volume medical centers, which should triage the patients who need to be transferred to higher referral centers for more aggressive treatment in an ICU.

### 5.7. Comparison of Scoring Systems

The Ranson score system was first developed to predict AP severity, followed by the Glasgow score, the modified version of the Ranson score. Subsequently, APACHE II, the BISAP score and JSS were developed. The accuracy of predicting severe AP was 0.81–0.88 when the Ranson score was 3 or more and 0.73–0.78 when the Glasgow score was 2 or more [60]. The AUC is 0.80–0.89 with an APACHE II score greater than seven points and 0.79–0.88 with a BISAP score greater than three points [60]. Some studies stated that the Glasgow score better predicts the severity of AP than APACHE II and the Ranson score. In contrast, other studies reported that the BISAP score shows a similar or superior accuracy in predicting the mortality compared with APACHE II and the Ranson score [61,62]. The comparison of scoring systems is summarized in Table 8.

## 6. Predictors Using Artificial Intelligence

There are two studies that compare the accuracy of machine learning (ML) models to APACHE II in predicting the severity of AP. In both studies, the ML model had a higher accuracy than APACHE II (AUC 0.92 vs. 0.63, 0.82 vs. 0.74) [63,64]. Mofidi et al. investigated the prediction of the severity of AP using clinical and laboratory findings in patients with AP, and reported a higher accuracy in severity, multi-organ failure and mortality prediction compared with APACHE II or the Glasgow score [65]. Halonen et al. reported that the artificial neural network model is more accurate in predicting the severity of AP than the Ranson score (AUC 0.847 vs. 0.655) but has no difference in accuracy compared with APACHE II (AUC 0.817) [66]. Likewise, there is a possibility for improvement in the prediction of the severity of AP using artificial intelligence models. However, algorithms should be standardized in artificial intelligence and validated through large-scale randomized trials.

## 7. Conclusions

Despite intense and various studies on the pathophysiology of AP, overall mortality rates have not significantly improved. This suggests that an early diagnosis and appropriate prediction of the severity of AP are of utmost importance. In the 40 years since the Ranson score system was first introduced, various scoring systems (such as the Glasgow score, APACHE, BISAP, JSS, and HAPS) have been suggested; however, to date, none can accurately predict severity at an early stage, are non-invasive, or easy to use in patients. Additionally, although CRP is widely used as a single biochemical marker, it has a poor ability to predict severity within 48 h of AP onset. There is also a lack of evidence regarding the use of procalcitonin and other cytokines to predict the severity in patients with AP. CT is an essential test for detecting pancreatic necrosis; however, necrosis cannot be observed within 48 h. 

In these limited environments, patients with AP should initially be assessed using the BISAP score, and other scoring systems can also be considered as supplementary applications. If patients with AP visit low-volume medical centers or hospitals without an ICU, HAPS can be applied to determine the requirement for transfer to higher referral centers.

In future, it is important that a consensus is reached in each society and gathered to develop an ideal scoring system for predicting and evaluating severity in patients with AP. Moreover, the evolution of new technologies for genetic, transcriptomic and proteomic profiles is expected to access the specific patterns of various pathophysiologic processes in AP through which novel biomarkers that accurately reflect patient conditions can be developed.

## Figures and Tables

**Table 1 medicina-58-00787-t001:** Revision of Atlanta classification for severity of acute pancreatitis.

Severity	Definitions
Mild	Absence of organ failure, absence of local complications
Moderately severe	Transient organ failure (within 48 h) and/or local * or systemic ^$^ complications
Severe	Persistent single or multiple organ failure (>48 h)

* local complications: peripancreatic fluid collections, pancreatic and peripancreatic necrosis (sterile or infected), pseudocyst and walled off necrosis (sterile or infected), gastric outlet dysfunction, splenic/portal vein thrombosis, necrosis of colon; ^$^ systemic complications: exacerbation of pre-existing co-morbidities, such as coronary artery disease or chronic lung disease, precipitated by acute pancreatitis.

**Table 2 medicina-58-00787-t002:** Computed tomography findings of acute pancreatitis according to Balthazar grade.

Grade	Computed Tomography Finding
A	Normal pancreas
B	Focal or diffuse enlargement of pancreas
C	Intrinsic pancreatic abnormalities with inflammatory changes in peripancreatic fat
D	Single peripancreatic fluid collection
E	Two or more fluid collections and/or air in retroperitoneal area (adjacent to pancreas)

**Table 3 medicina-58-00787-t003:** Computed tomography (CT) severity index and modified CT severity index.

	CT Severity Index		Modified CT Severity Index	
CT grade	Normal pancreas	0	Normal pancreas	0
	Focal or diffuse enlargement of pancreas	1	Pancreatic abnormalities with or without peripancreatic inflammation	2
	Intrinsic pancreatic abnormalities with inflammatory changes in peripancreatic fat	2	Pancreatic or peripancreatic fluid collection or fat necrosis	4
	Single peripancreatic fluid collection	3	-	
	Two or more fluid collections and/or air in retroperitoneal area (adjacent to pancreas).	4		
Pancreatic necrosis	None	0	None	0
	<30%	2	<30%	2
	30–50%	4	>30%	4
	>50%	6	-	
Extrapancreatic complications (one or more)	-		Pleural effusion, ascites, vascular complication (venous thrombosis, arterial hemorrhage), GI tract involvement (inflammation, perforation, intraluminal fluid collection), parenchymal complications (infarction, hemorrhage, fluid collection of subcapsular area)	2

**Table 4 medicina-58-00787-t004:** Ranson score in acute pancreatitis (non-biliary and biliary).

Ranson (Alcoholic or Others)	Ranson (Biliary)
At admission	At admission
Age > 55 years	Age > 70 years
WBC * > 16,000/mm^3^	WBC > 18,000/mm^3^
LDH ^$^ > 350 U/L	LDH > 400 U/L
AST ^#^ > 250 U/L	AST > 250 U/L
Glucose > 200 mg/dL	Glucose > 220 mg/dL
In initial 48 h	In initial 48 h
Hematocrit fall > 10%	Hematocrit fall > 10%
BUN ^¥^ increase > 5 mg/dL	BUN increase > 2 mg/dL
Calcium < 8 mg/dL	Calcium < 8 mg/dL
PaO_2_ < 60 mmHg	PaO_2_ < 60 mmHg
Base deficit > 4 mEq/L	Base deficit > 4 mEq/L
Fluid sequestration > 6 L	Fluid sequestration > 4 L
Each factor 1 point (total 0–11 points)	

* WBC, white blood cell; ^$^ LDH, lactate dehydrogenase; ^#^ AST, aspartate aminotransferase; ^¥^ BUN, blood urea nitrogen.

**Table 5 medicina-58-00787-t005:** Glasgow severity score for acute pancreatitis.

Age > 55 years
WBC * > 15,000/mm^3^
PaO_2_ < 60 mmHg
LDH ^$^ > 600 U/L
AST ^#^ > 200 U/L
Albumin < 3.2 g/dL
Calcium < 8 mg/dL
Glucose > 180 mg/dL
Urea > 45 mg/dL

* WBC, white blood cell; ^$^ LDH, lactate dehydrogenase; ^#^ AST, aspartate aminotransferase.

**Table 6 medicina-58-00787-t006:** Bedside index for severity in acute pancreatitis.

BUN * > 25 mg/dL
Impaired mental status (Glasgow Coma Scale Score < 15)
SIRS ^$^ (defined as two or more of the followings) (1) Body temperature < 26 °C or >38 °C (2) Respiratory rate > 20/min or PaCO_2_ < 32 mmHg (3) Pulse > 90/min (4) WBC ^#^ < 4000/mm^3^ or >12,000/mm^3^ or 10% immature bands
Age > 60 years
Pleural effusion detected on imaging
Each factor 1 point (total 0–5 points)

* BUN, blood urea nitrogen; ^$^ SIRS, systemic inflammatory response syndrome; ^#^ WBC, white blood cell.

**Table 7 medicina-58-00787-t007:** Japanese severity score for acute pancreatitis.

Factors	Clinical Findings	Laboratory Findings
Prognostic factor I (2 points per each finding)	Shock Impaired level of consciousness Respiratory failure Severe sepsis Disseminated intravascular coagulation	Base excess < −3 mEq/L Hematocrit < 30% after hydration BUN * > 40 mg/dL Creatinine > 2 mg/dL
Prognostic factor II (1 points per each finding)		Calcium < 7.5 mg/dL Glucose > 200 mg/dL Protein < 6.0 g/dL LDH ^$^ > 700 IU/L PaO_2_ < 60 mmHg (on room air) Prothrombin time > 15 s Platelet count < 10,000/mm^3^ Balthazar score D or E
Prognostic factor III	SIRS ^#^ score > 3 (2 points) Age > 70 years (1 point)	
- Stage 0, mild acute pancreatitis - Stage 1, moderate acute pancreatitis - Stage 2, severe acute pancreatitis I (severity score 2–8 points) - Stage 3, severe acute pancreatitis II (severity score 9–14 points) - Stage 4, extremely severe acute pancreatitis (severity score 15–27 points)		

* BUN, blood urea nitrogen; ^$^ LDH, lactate dehydrogenase; ^#^ SIRS, systemic inflammatory response syndrome.

**Table 8 medicina-58-00787-t008:** Comparison of scoring systems in prediction for severity of acute pancreatitis.

Scoring System	Cut-Off Value	AUC ^£^	Advantage	Disadvantage
Ranson score	3	0.810	Can predict not only severity but also organ failure and mortality	Requires 48 h for a complete evaluation
Glasgow score	2	0.780	Can predict mortality regardless of etiology	Requires 48 h for a complete evaluation
APACHE * II	7	0.895	Effective scoring system to predict severity	Parameters are complicated and over-estimates in old age
APACHE-O ^$^	7	0.893	Reflects APACHE II with obesity	Not superior to APACHE II
BISAP ^#^	3	0.875	Can predict not only severity but also organ failure and mortality	Good predictive system for mainly Western environment
JSS ^¥^	2	0.798	Similar accuracy to Ranson score	Parameters are complicated and focused on in-hospital mortality

* APACHE, Acute Physiology and Chronic Health Examination; ^$^ APACHE-O, Acute Physiology and Chronic Health Examination with obesity consideration; ^#^ BISAP, bedside index of severity in acute pancreatitis; ^¥^ JSS, Japanese severity scale; ^£^ AUC, area under the curve.

## Data Availability

Not applicable.

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
