# Peer review of "Predicting Severity of Acute Pancreatitis"

_medicina, 2022, doi:10.3390/medicina58060787_

Round 1

Reviewer 1 Report

Thank you for the oportunity to review the present paper, on an important subject. I would like to make the following suggestions :

  1. Significant language/text editing could help improve the quality of the paper.
  2. The authors stress at several points in the paper the importance of adequate prediction of severity. Please provide reasons to do so, especially when there is no adequate treatment that improves outcome.
  3. Consider adding a section on the prediction of harmless acute pancreatitis. this is also subject of relevant research related to the subject of this paper.
  4. Section 2 on patient factors: I miss the influence of triglicerides as a cause of pancreatitis and impact on severity. Also the statements in this section are rather blunt and could benefit from some nuancing. 
  5. Section 3.3 on CRP: please provide more data on sensitivity and specificity of CRP.
  6. Section 3.4. This section is rather descriptive. Please consider including the paper  PMID: 32938552  in the references and discuss the findings of this study.
  7. Section 4 on CT. Consider adding a section of additional CT findings predictive of severe disease, such as pleural effusion, body composition, muscle mass.
  8. Section 4.2 This section discusses EUS, abdominal ultrasound etc, but these modalities are not discussed with a clear focus on the subject of the paper: severity predicion. 
  9. Section 5.4 Please provide data on the sensitivity / specificity.
  10. Consider using a summary table with severity predictors with pro's/con's and gross estimate of ROC or sensitivity/specificity.
  11. I miss a section where the authors not only describe, but also summarize current literature, and advice the reader on how to (A) move forward scientifically and (B) use all the options for their next patient. Give the reader more guidance than is currently provided.
  12. line 219: why per country? That seems to be not rational?

In summary, I think the paper could be markedly improved with some extra effort.

Author Response

Thank you for your nice and detailed reviews.

I did my best to revise my manuscript as you mentioned.

I attached the response to your comments and make a yellow-highlight in revised manuscript.

Reviewer 2 Report

I read with great interest the Review by Lee & Cho on predicting severity in acute pancreatitis (AP). The authors give a very informative overview of different patient-related factors to determine the severity of AP. Therefore, they discuss clinical, biochemical and radiologically factors as well as different scoring systems derived from these factors. The authors point to the difficulty to determine the extend of severity in the early acute pancreatitis in which almost every score is unprecise. In addition, the authors point to the need for additional biochemical markers in the initial phase of AP which should be investigated in future studies. 

I have just some minor comments for the athors:

  • The last sentence of the abstract is incomplete: ... and limitations is presented.
  • The sentence in line 45: what is past "medical history taking"?
  • The first sentence in line 58 is incomplete
  • Please add a reference in line 87. 
  • In line 103: You should first introduce APACH II score 
  • In line 136: Please indicate why EUS is useful in the detection of the etiology of AP -> biliary AP?  

Author Response

Thank you for your nice and detailed reviews.

I did my best to revise my manuscript as you mentioned.

I attached the response to your comments and make a green-highlight in revised manuscript.

Round 2

Reviewer 1 Report

the authors have adequately adressed previous concerns, I feel the present paper is suitable for publication

Author Response

the authors have adequately adressed previous concerns, I feel the present paper is suitable for publication

--> This revision work was a good  opportunity for me to learn a lot. Thank you for your good comments.  

This manuscript is a resubmission of an earlier submission. The following is a list of the peer review reports and author responses from that submission.